# Low prevalence of helminth infection in Ugandan children hospitalized with severe malaria

Drew Capone[1]*, Nuzrath Jahan[1], Ruth Namazzi[2,3], Robert O. Opoka[3,4], Chandy C. John[5]

1 Department of Environmental and Occupational Health, School of Public Health, Indiana University, Bloomington, Indiana, United States of America, 2 Makerere University, College of Health Sciences, Kampala, Uganda, 3 Global Health Uganda, Kampala, Uganda, 4 Aga Khan University East Africa, Nairobi, Kenya, 5 Department of Pediatrics, School of Medicine, Indiana University, Indianapolis, Indiana, United States of America

* dscapone@iu.edu

## Abstract

Co-infection by intestinal helminths and *Plasmodium* spp. may be common in endemic communities. In 2003, Uganda instituted a national deworming program, with anti-helminth medication provided twice annually to children 6 months to 5 years of age, but few follow-up studies have been conducted. Several studies have identified a relationship between helminth infection, *Plasmodium* spp. infection and malaria severity. However, the relationship is not well defined, and results are inconclusive. We analyzed 177 stool samples from a cohort of children with severe malaria enrolled in two hospitals in Uganda from 2014–2017. All children were 6 months to 48 months of age and had a clinical presentation of and laboratory confirmation for severe malaria. We also analyzed 25 stool samples from community children who were negative for malaria via rapid diagnostic test and were enrolled from the same household or neighborhood and matched by age, sex, and time of enrollment. We investigated if intestinal helminth infection modified risk of severe malaria. We extracted nucleic acids from stool and tested them for six helminth species (*Anyclostoma duodenale*, *Ascaris lumbricoides*, *Necator americanus*, *Strongyloides stercolaris*, *Trichuris trichiura*, *Shistosoma mansoni*) using highly sensitive quantitative PCR. We found a low prevalence of infection by ≥1 intestinal helminth species in children with severe malaria (5.1%, n = 9/177) and community control children (4.0%, n = 1/25). Helminth infection did not increase or decrease the risk of severe malaria in this cohort (aRR = 1.0, 95% Confidence Interval = 0.82, 1.3, p = 0.78). In these areas of Uganda, the national deworming campaign has been highly successful, as stool-based helminth infection was rare even when using sensitive methods of detection and helminths were not associated with severe malaria in this study.

**Data availability statement:** All relevant data are within the paper and its Supporting Information files.

**Funding:** This study was funded by the United States National Institutes of Health and was awarded to CJ (NINDS, R01 NS055349, Fogarty International Center, D43TW010928).

**Competing interests:** The authors have declared that no competing interests exist.

## Introduction

Soil-transmitted helminths (STH) are parasitic worms that live in the human gut during the adult stage of their lifecycle. Worms lay eggs, which are shed in human feces at high concentrations (e.g., up to 500,00 eggs per gram feces) [1]. Following a period of incubation in soil, transmission occurs when people ingest or contact the infective lifecycle stage. The most common STHs include roundworms (*Ascaris lumbricoides*), whipworms (*Trichuris trichiura*), and hookworms (*Necator americanus* and *Ancylostoma duodenale*). The Global Burden of Disease study estimates that 294 million people are infected by *Ascaris*, 267 million by *Trichuris*, and 113 million by hookworm (2021 data) [2]. *Strongyloides stercolaris* is a rare STH species but is unique in that autoinfection is possible and persistent infections have been reported for decades [3]. *Schistosoma mansoni* is another intestinal helminth that differs from STHs because the *Schistosoma* lifecycle requires development in snails in freshwater, where exposure occurs. It is estimated that *Schistosoma* spp. infect 151 million people (2021 data) [2].

The burden of disease from intestinal worms is concentrated in low- and middle-income countries (LMIC) without adequate sanitation [4]. Yet, modelling in endemic communities has demonstrated that persistent helminth transmission is possible even in areas where sanitation coverage is high [5]. As an integrated approach to helminth control, the World Health Organization recommends improving sanitation infrastructure as well as hygiene education and preventative chemotherapy [6]. Preventive chemotherapy for intestinal helminths involves the periodic administration of anthelmintic drugs to at-risk populations. As a part of preventative chemotherapy efforts, mass drug administration (MDA) programs are employed in endemic areas where these parasitic infections are prevalent [7]. These programs aim to treat entire populations, regardless of infection status, with a single dose or multiple doses of anthelmintic drugs such as albendazole or mebendazole, depending on the specific parasites present in the region.

In Uganda, a nationwide survey of more than 20,000 school children was conducted for STHs from 1998–2005 [8]. Approximately half of children (55%) were infected by ≥1 STH. The prevalence of *Trichuris trichiura* was 5.0% (range: 0–68% by school), *Ascaris lumbricoides* was 6.3% (0–89% by school), and hookworm was 44% (0–90% by school). In response to the high prevalence of helminth infection, the Uganda Ministry of Health began implementation of its national STH control program in 2003, including twice yearly deworming for school age children [9]. Later, in 2005, Uganda began implementation of biannual "Child Health Days". These health fairs are campaign style events organized by District Health Teams, which includes deworming every six months for children 12–60 months of age [10].

Malaria is an infectious disease caused by *Plasmodium* spp. parasites, which are transmitted to humans through the bites of an infected female *Anopheles* mosquito. After transmission, *Plasmodium* enters the bloodstream and invades red blood cells, where the parasite replicates. Most cases of malaria are uncomplicated or even asymptomatic [11]. Severe malaria occurs in a small subset of cases, and is

characterized by complications that include prostration, severe anemia, multiple seizures, respiratory distress, or coma. The World Health Organization estimated there were 249 million cases of malaria in 2023, a 7% increase from 2019 [12].

The 85 countries where malaria is endemic have substantial geographic overlap with countries where STHs are endemic [13,14]. While co-infection is common, there is mixed evidence regarding the association between *Plasmodium* spp. infection, malaria severity, and STH infection. Individual studies on co-infections have reported conflicted findings [15]. A 2016 meta-analysis found that STH infection may increase susceptibility to uncomplicated or asymptomatic *P. falciparum* infection but may protect against malaria-induced anaemia [16]. A 2021 meta-analysis found in the unadjusted analysis that STH infection was protective against *P. falciparum* infection, but after adjusting for potential confounders observed that STH infection may increase susceptibility to *P. falciparum* infection [15].

Helminth infection often modulates host immunity by shifting immune response towards a Th2-dominant profile and inducing regulatory pathways that suppresses inflammation [17,18]. This presents two contrasting possibilities. Immune modulation may limit an effective Th1-mediated response necessary to control infection by *P. falciparum*, while helminth-driven immunoregulation may mitigate immunopathology associated with severe malaria. Empirical data on helminth and *P. falciparum* infection may provide further insight into these interactions.

The Neurodevelopment Outcomes in Children with Severe Malaria Study (NDI) enrolled children with severe malaria and asymptomatic community control children from the same neighborhood or household in Kampala and Jinja, Uganda from 2014–2017. We tested stool samples from a subset of these study children to determine if intestinal helminth infection modified risk of severe malaria among children enrolled in the study. Our research aims were to (1) assess the prevalence of common intestinal helminth species among young children enrolled in this study using highly sensitive molecular methods; and (2) determine if intestinal helminths modified risk of severe malaria in hospitalized pediatric patients compared to community controls enrolled in this study.

## Methods

### NDI study

The primary outcome of the NDI study was to assess cognitive outcomes in children 12-months following hospitalization with severe malaria compared to matched controls. We prospectively enrolled 600 children with severe malaria and 120 community children 6 months to 48 months of age at Mulago National Referral Hospital in Kampala and Jinja Regional Referral Hospital in Jinja, Uganda from 28/03/2014–17/04/2017 [19,20]. The 600 children with severe malaria included five groups of 120 children corresponding to each of the five most common types of severe malaria (i.e., cerebral malaria, respiratory distress, malaria with complicated seizures, severe malarial anemia, and prostration) [19,20]. Full details of study enrollment, inclusion and exclusion criteria and study procedures have been previously published [19,20]. Eligibility criteria for children with severe malaria included evidence for malaria parasitemia via a rapid diagnostic test or by Giemsa microscopy and ≥1 of the five common severe malaria features in Uganda (prostration, severe anemia, multiple seizures, respiratory distress, and coma). Community children were enrolled from biological siblings or were neighborhood-matched to enrolled children with severe malaria. Controls were enrolled to be within the same age range as the children with severe malaria. Inclusion criteria for community controls were residence in the same or nearby neighborhood as an enrolled child with severe malaria and being 6 months to 48 months of age. Community control children were tested for asymptomatic *Plasmodium* infection via a rapid diagnostic test (RDT). Exclusion criteria included an illness requiring medical care within the previous four weeks, a major medical or neurological abnormality at screening physical examination, or an active illness or axillary temperature on screening of >37.5°C.

Following admission, child feces were collected from the child's first bowel movement into sterile collection containers, which were aliquoted into 2mL cryovials. Community children were provided a stool container and asked to bring back a stool sample within 1 month of enrollment. Samples were stored at -80C in Uganda shipped to Indiana University on dry ice at the completion of the study. Samples at Indiana University were again stored at -80C.

## Fecal analysis for helminths

We randomly selected a subset of 202 stools from the 502 samples available for inclusion in this study based on the following criteria: (1) the stool sample was collected ≤48 hours after admission (only for children with severe malaria); (2) adequate stool was available for nucleic acid extraction (i.e., ≥100 mg); and (3) community control children were negative for *Plasmodium* infection via RDT. These criteria included fecal samples from 177 children with severe malaria and 25 community controls.

Among the 177 children with severe malaria, 16 presented with cerebral malaria, 39 with respiratory distress, 60 with multiple seizures, 32 with severe malaria anemia, and 30 with prostration. Among all severe malaria children in the NDI study, 7.3% died. However, mortality was less frequent among the children (2.3%, n = 4/177) included in this study because we were not able to obtain fecal samples from many of the children who died.

A sample size of 177 children with severe malaria and 25 community control children provided ≥80% power to detect a ≥17% absolute difference in intestinal helminth infection rates in the children with severe malaria, if the baseline rate of stool helminth infection in the control group was 20% [8,21]. We extracted DNA from 100 mg of child feces using the QIAamp 96 Virus QIAcube HT Kit (Qiagen, Hilden, Germany), which we automated on the QIAcube. Extraction included a pre-treatment step with Qiagen PowerBead Pro Tubes (Qiagen, Hilden, Germany), which we have previously validated for multi-pathogen analysis [22,23]. We included at least one negative extraction control on each day of DNA extractions and spiked an extraction positive control (i.e., bovine herpes virus) into each sample.

We used quantitative PCR (qPCR) to measure nucleic acids for five common STH species including *Anyclostoma duodenale*, *Ascaris lumbricoides*, *Necator americanus*, *Strongyloides stercolaris*, and *Trichuris trichiura* as well as *Shistosoma mansoni*, which is another intestinal helminth species endemic in Uganda [24]. PCR was performed via a custom TaqMan Array Card (TAC). We mixed 40 µL of nucleic acid template with 60 µL of AgPath-ID™ One-Step RT-PCR Reagents, which was added to each port on the TAC. The TAC was centrifuged at 1200 RPM for minute twice, sealed, the loading port trimmed off, and then the TAC was analyzed on a Quantstudio 7 Flex Instrument (Applied Biosystems, Waltham, MA) using the following cycling conditions: 45°C for 20 min and 95°C for 10 min, followed by 45 cycles of 95°C for 15 s and 60°C for 1 min. At least one positive control (i.e., a plasmid with all gene targets) and one negative control was run each day of PCR analysis. We developed a standard curve for TAC using the methods described in Kodani and Winchell 2012 [25]. Quantification cycle was determined via manual thresholding, by comparing sample amplification against our daily positive and negative controls. Additional data on PCR reaction conditions and QA/QC can be found in the appendix (S1 Text, S1 Table).

We generated summary statistics using Microsoft Excel 365 and performed statistical analyses in RStudio (R Foundation for Statistical Computing, Vienna, Austria, Version 4.2.3). We used generalized estimating equations (geepack package) to fit Poisson regression models with robust standard errors and an exchangeable correlation structure [26]. We assessed if infection by ≥1 helminth modified risk of severe malaria [27]. In our model, severe malaria status was the dependent variable, while helminth infection (binary), age (categorical: <12 month, 13–24 months, >24 months), and socioeconomic status (quantile) were included as independent variables. We determined household socioeconomic status using a previously validated scoring instrument that assessed material possessions, house structure, and access to food, water, and electricity [28]. We accounted for clustering by study site [29].

## Ethical approval

Initial verbal consent from the parents or legal guardians of study participants was obtained for children fulfilling inclusion criteria, since most participants were critically ill and required emergency stabilization. Written informed consent was obtained once the participant was clinically stabilized. Ethical approval was granted by the Institutional Review Boards at Makerere University School of Medicine, the University of Minnesota, Indiana University, and regulatory approval granted by the Uganda National Council for Science and Technology.

## Results

We found a low prevalence of helminth infection among children with severe malaria and community controls. The prevalence of any helminth infection was similar in children with severe malaria (5.1%, n = 9/177) and community control children (4.0%, n = 1/25) (Table 1). *Necator americanus* was the most frequently detected helminth (severe malaria 1.7%, community controls 4.0%), followed by *Trichuris trichiura* (severe malaria 1.7%, community controls 0%), *Shistosoma mansoni* (severe malaria 1.1%, community controls 0%), *Ascaris lumbricoides* (severe malaria 0.6%, community controls 0%), and we did not detect *Strongyloides stercolaris* or *Ancylostoma duodenale*. Infection by ≥1 of the helminths assessed was not associated with severe malaria (aRR = 1.0, 95% Confidence Interval = 0.82, 1.3, p = 0.78) via Poisson regression modelling (S2 Table).

Helminth prevalence was similar among children enrolled in Kampala (6.7%, n = 7/105) and Jinja (3.1%, n = 3/97) (t-test p value = 0.24). The single *Ascaris* infection and both *Schistosoma* infections were detected in children from Kampala. Hookworm infections (i.e., *Necator americanus*) were split equally across the sites (n = 2 in Kampala and n = 2 in Jinja). For *Trichuris* two of the three infections were in children from Kampala while the third infection was observed in Jinja.

## Discussion

We observed a low prevalence of intestinal helminth infection in children with severe malaria and community controls who were enrolled in this study. These results support the effectiveness of the national semi-annual deworming campaign in Uganda. In addition, our results add to previous studies whiched showed a decrease in intestinal helminth infection in Uganda via Kato-Katz method [30] because our results demonstrate that infections were uncommon in these communities even when stool was tested by highly sensitive qPCR, which is more sensitive than microscopy [31]. Due to the low prevalence of intestinal helminth infection, and especially the low prevalence of individual species, these results do not offer insight into the relationship between helminths and the susceptibility to *Plasmodium* spp. infection or malaria severity among young children. Instead, our results provide evidence that severe malaria occurred frequently in an area with a very low prevalence of intestinal helminth infections, as evaluated by qPCR. The study data suggest that in these areas, intestinal helminth infections play little role in the risk of development of severe malaria.

While intestinal helminth infection prevalence was low, other factors may modify risk of *Plasmodium* spp. infection and malaria severity. Infection by pathogenic enteric viruses, bacteria, and protozoa are common in children in low-income countries, with combined enteric pathogen prevalence often approaching 100% [32–34]. It remains unclear if and how other enteric pathogens may modify risk of *Plasmodium* infection and malaria severity. However, there is increasing

**Table 1. Prevalence of intestinal helminth infection among children with severe malaria and community controls.**

| Pathogen | Severe Malaria | | Community Controls | |
|---|---|---|---|---|
| | Prevalence | Median gene copies/gram feces)* | Prevalence | Median gene copies/gram feces* |
| ≥1 helminth | 5.1% (9/177) | | 4.0% (1/25) | |
| *Anyclostoma duodenale* | 0% (0/177) | Not detected | 0% (0/25) | Not detected |
| *Strongyloides stercolaris* | 0% (0/177) | Not detected | 0% (0/25) | Not detected |
| *Ascaris lumbricoides* | 0.6% (1/177) | 3.1 log10 | 0% (0/25) | Not detected |
| *Schistosoma mansoni* | 1.1% (2/177) | 6.5 log10 | 0% (0/25) | Not detected |
| *Trichuris trichiura* | 1.7% (3/177) | 2.3 log10 | 0% (0/25) | Not detected |
| *Necator americanus* | 1.7% (3/177) | 5.1 log10 | 4.0% (1/25) | 2.6 log10 |

Note: *Calculation of median excludes non-detects.

evidence the gut microbiome is a risk factor of severe malaria and that the gut microbiome plays a role in malaria pathogenesis [35].

National level surveillance for intestinal helminth infections has not been conducted in Uganda since the 1998–2005 nationwide survey [8]. A follow-up study of five districts in 2022 – which did not overlap with the districts we enrolled children from – found that intestinal helminth prevalence had decreased in most districts, but localized heterogeneity in infection prevalence persists [30]. Another study of six regions of Uganda, which also did not include Kampala and Jinja in its testing, found low rates of stool helminth infection, with some heterogeneity by region, but overall rates, particularly for hookworm infection (*Necator americanus* or *Ancylostoma duodenale*) were higher than in this study. Of note, all these studies used Kato-Katz microscopy to detect stool helminth infections, while in the current study we used qPCR to detect intestinal helminth ova shed in stool. qPCR has repeatedly been shown to be more sensitive than microscopy for diagnosing helminth infections, particularly in cases of low-intensity infections [36–38]. Thus, the results of this study provide strong new evidence for lack of stool helminth infection, even at very low concentrations, in most children in Kampala and Jinja.

The World Health Organization has shifted its focus from coverage-based deworming targets to eliminating helminth morbidity as a public health problem. Elimination of helminth morbidity is defined in this program as <2% prevalence in children of moderate-to-heavy intensity infection from any STH [39]. However, there remains a lack of consensus on interpreting qPCR data to determine infection severity because the number of gene copies per ova varies between species and increases as the ova develops [40]. For example, the number of gene copies of the *Ascaris* spp. gene target ITS-1 increases from <10 per egg to approximately 10,000 after 10 days of incubation [41]. Though our study had a small sample size and did not characterize infection severity, the low prevalence of any helminth infection we detected provides suggestive evidence that the WHO morbidity target is close to being achieved in the communities we studied. It is possible that some de-worming resources could be re-allocated from the districts we studied to other districts in Uganda where intestinal helminths remain a greater public health problem, though this could risk a recurrence of higher rates of intestinal helminth infection in this area.

Intestinal helminth surveys in Uganda – as well as in other endemic countries – primarily use microscopy (e.g., kato-katz, mini-FLOTAC) to detect eggs in stool [8,42]. Molecular techniques, such as qPCR, offer superior sensitivity to microscopy and may be advantageous for helminth studies in low-prevalence communities [43]. While microscopy is labor-intensive and requires highly trained technicians, the reagents and equipment needed for microscopy are substantially cheaper than for molecular methods. In low- and middle-income countries where helminths are endemic, molecular methods may be a useful complementary tool to microscopy for surveillance in communities with a low prevalence of infection.

Study limitations include a small sample size, which permitted detection of only large differences in stool helminth infection prevalences between children with severe malaria and community children. Our study was underpowered to differentiate between small differences in prevalences, as we observed here. Further, mortality prevalence was lower among severe malaria children in this study compared to the overall NDI study. Our analysis may have missed some of the high severity cases. In addition, this study's sample size was much smaller than those of helminth surveillance studies in Uganda [30]. Study strengths include the rigorously characterized cohort of children with severe malaria and the use of modern molecular methods with a higher sensitivity than microscopy [43]. Future work on the relationship between intestinal helminth infection and malaria in communities with a higher prevalence of helminth infection will be important to better elucidate the relationship between these pathogens. However, the study results suggest that in Kampala and Jinja, stool helminth infection is rare and therefore an unlikely contributor to risk of severe malaria.

Community helminth and malaria prevalence are mediated by multiple factors including the environment, household infrastructure, personal behavior (e.g., wearing shoes to prevent hookworm infection or using insecticide treated bed nets to prevent malaria), and the frequency of preventative chemotherapy [1,6,44–46]. Integrated approaches – such as

those promoted by the World Health Organization – have made substantial progress in reducing morbidity from helminth infections globally since the year 2000 [2]. Control efforts have also dramatically reduced mortality and morbidity due to malaria in the same period, but climate change, parasite resistance to frontline medications and insecticides, and humanitarian crises threaten to derail progress [12]. While evidence for the relationship between helminths and malaria remains mixed, there is a clear and urgent need to continue integrated efforts towards reducing the global burden of disease from these parasites Our findings do not support a role for intestinal helminth infections in modifying the risk of severe malaria in young children in Kampala and Jinja. This suggests that in similar low-prevalence settings, intestinal helminths may be unlikely to contribute to the pathogenesis of severe malaria.

## Supporting information

**S1 Text. Molecular methods.**
(DOCX)

**S1 Table. Assay performances.**
(DOCX)

**S2 Table. Regression analysis.**
(DOCX)

## Acknowledgments

The authors thank the study participants and caregivers for their participation in this study, and the study team for their work in enrollment, care, testing and follow-up of study participants.

## Author contributions

**Conceptualization:** Drew Capone, Ruth Namazzi, Robert O. Opoka, Chandy C. John.

**Data curation:** Drew Capone, Nuzrath Jahan.

**Formal analysis:** Drew Capone.

**Funding acquisition:** Ruth Namazzi, Robert O. Opoka, Chandy C. John.

**Investigation:** Nuzrath Jahan, Robert O. Opoka, Chandy C. John.

**Methodology:** Drew Capone, Chandy C. John.

**Writing – original draft:** Drew Capone.

**Writing – review & editing:** Chandy C. John.

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
