## [Decision Letter · Decision Letter 0]

22 May 2025

Dear Dr. Capone,

**Please respond in detail especially to the extensive comments from Reviewer #2.**

We look forward to receiving your revised manuscript.

Kind regards,

David J. Diemert, M.D.

Academic Editor

PLOS ONE

**Journal Requirements:**

1. When submitting your revision, we need you to address these additional requirements. Please ensure that your manuscript meets PLOS ONE's style requirements, including those for file naming. The PLOS ONE style templates can be found at https://journals.plos.org/plosone/s/file?id=wjVg/PLOSOne_formatting_sample_main_body.pdf and https://journals.plos.org/plosone/s/file?id=ba62/PLOSOne_formatting_sample_title_authors_affiliations.pdf 2. Please include a complete copy of PLOS’ questionnaire on inclusivity in global research in your revised manuscript. Our policy for research in this area aims to improve transparency in the reporting of research performed outside of researchers’ own country or community. The policy applies to researchers who have travelled to a different country to conduct research, research with Indigenous populations or their lands, and research on cultural artefacts. The questionnaire can also be requested at the journal’s discretion for any other submissions, even if these conditions are not met. Please find more information on the policy and a link to download a blank copy of the questionnaire here: https://journals.plos.org/plosone/s/best-practices-in-research-reporting. Please upload a completed version of your questionnaire as Supporting Information when you resubmit your manuscript. 3. Thank you for stating the following financial disclosure: This study was funded by the United States National Institutes of Health and was awarded to CJ (NINDS, R01 NS055349, Fogarty International Center, D43TW010928).  Please state what role the funders took in the study.  If the funders had no role, please state: "The funders had no role in study design, data collection and analysis, decision to publish, or preparation of the manuscript." If this statement is not correct you must amend it as needed. Please include this amended Role of Funder statement in your cover letter; we will change the online submission form on your behalf.

Reviewers' comments:

Reviewer's Responses to Questions

**Comments to the Author**

1. Is the manuscript technically sound, and do the data support the conclusions?

Reviewer #1: Yes

Reviewer #2: Partly

2. Has the statistical analysis been performed appropriately and rigorously?

Reviewer #1: Yes

Reviewer #2: No

3. Have the authors made all data underlying the findings in their manuscript fully available?

Reviewer #1: Yes

Reviewer #2: Yes

4. Is the manuscript presented in an intelligible fashion and written in standard English?

Reviewer #1: Yes

Reviewer #2: No

**Reviewer #1: ** This is an automated report for PONE-D-25-08603. This report was solicited by the PLOS One editorial team and provided by ScreenIT.

ScreenIT is an independent group of scientists developing automated tools that analyze academic papers. A set of automated tools screened your submitted manuscript and provided the report below. Each tool was created by your academic colleagues with the goal of helping authors. The tools look for factors that are important for transparency, rigor and reproducibility, and we hope that the report might help you to improve reporting in your manuscript. Within the report you will find links to more information about the items that the tools check. These links include helpful papers, websites, or videos that explain why the item is important. While our screening tools aim to improve and maintain quality standards they may, on occasion, miss nuances specific to your study type or flag something incorrectly. Each tool has limitations that are described on the ScreenIT website. The tools screen the main file for the paper; they are not able to screen supplements stored in separate files. Please note that the Academic Editor had access to these comments while making a decision on your manuscript. The Academic Editor may ask that issues flagged in this report be addressed. If you would like to provide feedback on the ScreenIT tool, please email the team at ScreenIt@bih-charite.de. If you have questions or concerns about the review process, please contact the PLOS One office at plosone@plos.org.

**Reviewer #2:**  All detailed reviewer comments are provided in the attached PDF. While the study addresses an important question, key aspects need clarification. Study design and methods lack clarity; statistical tests are underpowered or unjustified. Conclusions don’t clearly address the aim or interpret the lack of association between helminths and severe malaria.

**Do you want your identity to be public for this peer review?** For information about this choice, including consent withdrawal, please see our Privacy Policy

Reviewer #1: No

Reviewer #2: No

---

## [Author Response · Author response to Decision Letter 1]

23 Jul 2025

PLOS One,

We have attached a point-by-point response to the reviewer's comments.

I have included a completed inclusivity in global research questionnaire.

I confirm that my submission contains all raw data required to replicate the results of your study.

Regards,

Drew Capone

---

## [Editor Report · Decision Letter 1]

28 Aug 2025

Low prevalence of helminth infection in Ugandan children hospitalized with severe malaria

PONE-D-25-08603R1

Dear Dr. Capone,

We’re pleased to inform you that your manuscript has been judged scientifically suitable for publication and will be formally accepted for publication once it meets all outstanding technical requirements.

Kind regards,

David J. Diemert, M.D.

Academic Editor

PLOS ONE
---

## [Editor Report · Acceptance letter]

PONE-D-25-08603R1

PLOS ONE

Dear Dr. Capone,

I'm pleased to inform you that your manuscript has been deemed suitable for publication in PLOS ONE. Congratulations! Your manuscript is now being handed over to our production team.

Kind regards,

on behalf of

Dr. David J. Diemert

Academic Editor

PLOS ONE